# Elucidating The Impact of Community-Level Social Determinants of Health on Pre-operative Frailty: A Data-Driven Study in Florida

Chen Bai
*Department of Health Outcomes and*
*Biomedical Informatics*
*University of Florida*
Gainesville, FL, USA
chenbai@ufl.edu

Mamoun Mardini
*Department of Health Outcomes and*
*Biomedical Informatics*
*University of Florida*
Gainesville, FL, USA
malmardini@ufl.edu

*Abstract*—*Frailty, an age-related syndrome, is associated with poor post-operative outcomes. The impact of community-level social determinants of health (SDoH) on pre-operative frailty has not been investigated yet. We developed a machine learning model to predict pre-operative frailty using an institutional dataset and applied it to a more geographically diverse population from the OneFlorida+ Clinical Research Consortium. Computable phenotyping for SDoH stratification using unsupervised learning was employed to identify distinct patient profiles based on zip code-level SDoH characteristics. We applied multivariate logistic regression to examine the association between SDoH profiles and pre-operative frailty risk. Adverse community-level SDoH profiles are independently associated with higher pre-operative frailty risk; patients from the disadvantaged SDoH profile had 1.21 times higher odds (95% CI 1.16-1.26) of being frail compared to the advantaged SDoH cluster after adjusting for potential confounders. Considering patients' social context could improve pre-operative care and surgical outcomes, informing clinical practice and policies.*

**Keywords—Clustering, frailty, machine learning, preoperative, social determinant of health**

## Introduction

The rate of global population aging has accelerated more markedly than in the past. According to projections by the World Health Organization in 2022, the number of individuals aged 60 years and older will double to reach 2.1 billion worldwide by 2050 [1]. This significant demographic shift underscores the importance of healthcare providers to understand the determinants that make older adults vulnerable to deteriorations in health and functionality. Frailty, an age-related syndrome characterized by decreased physiological reserve and vulnerability, is one of the primary determinants. The prevalence of frailty rises with age [2], [3], and frail older adults are at substantially increased risk of poor post-operative outcomes, including mortality, disability, falls, and hospitalization [4], [5], [6], [7], [8], [9].

Environmental factors are crucial in unraveling the complexities of the aging process, which is fundamental to understanding the development of frailty[10]. Social determinants of health (SDoH) are an important set of environmental factors defined as conditions where individuals are born, live, work, and age that shape health outcomes, including income, education, housing, and access to healthcare, etc. [11] SDoH have emerged as important factors influencing individuals' health outcomes across various domains. Numerous studies have investigated the impact of SDoH on various health outcomes such as diabetes, heart failure, and pre-mature mortality [12], [13], [14]. To better understand the factors contributing to pre-operative frailty, it is essential to take the broader social and environmental context in which individuals live into consideration.

Previous studies have primarily focused on the association between individual-level factors, such as socioeconomic status and functional status with frailty [15], [16]. However, the influence of community-level SDoH, which reflects the broader social and environmental context where individuals live and work has not been adequately addressed, specifically in pre-operative settings. Community-level SDoH, such as neighborhood deprivation and access to healthcare services may have a significant impact on individuals' frailty. For example, Lang et al. found that frailty is independently associated with both individual and neighborhood socioeconomic factors, with the mean frailty index of individuals living in the most deprived neighborhood being nearly twice that of those living in the least deprived neighborhood [17]. Understanding the relationship between community-level SDoH and pre-operative frailty is essential for healthcare providers and policymakers to develop comprehensive strategies to promote post-surgical outcomes in vulnerable populations.

This study aims to investigate the impact of community-level SDoH on pre-operative frailty in a diverse population of surgical patients in Florida. We built a machine learning model to predict pre-operative frailty using an institutional electronic health records (EHR) data source. We applied the model to a larger EHR database to obtain the pre-operative frailty status for a more geographically diverse population. Computable phenotyping for SDoH stratification using unsupervised learning was employed to identify distinct patient profiles based on zip code-level (5-digit) SDoH characteristics. By leveraging a large-scale and data-driven approach, we seek to uncover the complex interplay between community-level SDoH and pre-operative frailty.

## Methods

### A. Data Sources

In this study, we developed a pre-operative frailty prediction model using de-identified data from 14,000 patients collected at the University of Florida (UF IDR) from January 2018 to December 2019 as part of a federally funded

study approved by the UF institutional review board (IRB). To extend the diversity of patients' living situations and socioeconomic status, we applied the prediction model to patients from the OneFlorida+ Clinical Research Consortium (OneFL+). OneFL+ consists of electronic health records linked to various other data sources for about 16.8 million Floridians since 2012. Therefore, patients from OneFL+ represent a broad cross-section of communities in Florida, allowing us to capture more diverse community-level SDoH.

### B. Fried Frailty Phenotype

Fried frailty phenotype was assessed in the pre-operative clinic and obtained in UF IDR data. Patients were labeled as frail if they exhibited/reported greater than or equal to three of the following: (1) unintended weight loss of ≥ 10 pounds within the last six months; (2) subjective exhaustion, defined as endorsing moderate feelings that everything they did was an effort over the last week or moderate feelings that they could not "get going" in the last week; (3) slow walking speed, determined by nurses' observation; (4) weak grip strength; defined according to gender-specific T-score of -2.5 on the maximum grip strength measurement from the three trials using Jamar hydraulic hand dynamometer (Model J00105, Lafayette Instrument Europe, Leicester, UK); and (5) self-reported low physical activity as defined by the Duke Activity Status Index [18]. The outcome was scored 0-5, where 0-2 = non-frail and 3-5 = frail. We used these labels as our ground truth to train and evaluate our prediction model.

### C. Pre-operative Frailty Prediction using UF IDR Data

Because frailty is not consistently assessed in clinical settings and not captured in EHRs such as OneFL+, we developed a pre-operative frailty prediction model using UF IDR data which used Fried frailty phenotype as the ground truth. We subsequently applied the developed model to predict pre-operative frailty status for patients in OneFL+ cohort. We excluded patients with more than two missing frailty components from UF IDR data, resulting in 8,999 patients for training the prediction model. Features included patients' characteristics that were available up to 6 months before their frailty assessments: socio-demographic (e.g., age and gender), medical history and acuity (e.g., history of diagnoses), and the most recent biochemical measurements that are commonly measured in pre-operative clinics (e.g., hemoglobin and hematocrit). We applied eXtreme Gradient Boosting (XGBoost) to build the machine learning model. To overcome data imbalance, the weights of the minority class were set to the ratio between the number of non-frail and frail patients. We used nested cross-validation with five outer folds and five inner folds to evaluate the machine learning model. Detailed information regarding the model development and evaluation can be found in our previous paper [19].

### D. Determining the Pre-operative Frailty Status in OneFL+ Cohort

We extracted the most recent surgical encounters for patients in OneFL+ from 2012 to 2022. To ensure sufficient medical history, we excluded patients with fewer than two medical encounters in the six months prior to the targeted surgery. Using the same features that were utilized to develop the pre-operative frailty prediction model with UF IDR data, we extracted a parallel set of features from the OneFL+ cohort. We then applied the previously trained prediction model to this OneFL+ cohort to obtain the predicted pre-operative frailty status for these patients. We selected a probability cut-point of 0.8 to categorize predicted frail and non-frail patients in the OneFL+ cohort, where patients with a predicted frailty probability exceeding 0.8 were classified as frail.

### E. Community-level Social Determinant of Health

We incorporated zip code-level SDoH from two data sources: (1) the American Community Survey (ACS) conducted by the US Census Bureau and (2) the Population Level Analysis and Community Estimates (PLACES) data from the Centers for Disease Control and Prevention (CDC). The ACS data offered SDoH information such as demographics, housing, transportation, and other socioeconomic factors. The PLACES data provided statistics on a wide range of population characteristics, including their health behaviors and health outcomes. We included four broad categories of zip code-level SDoH information, consisting of 37 features in total from the ACS as described in Table 1. A total of 18 features from PLACES data were included, resulting in 55 SDoH features in total.

TABLE I.      ZIP CODE-LEVEL SDoH FEATURES FROM THE ACS DATA AND DETAILED DESCRIPTIONS.

| | SDoH Category | Number of Variables | Variable Descriptions |
|---|---|---|---|
| **Area Demographics** | Living situation | 4 | % of population living alone; average household; % of population over 65 who do not have telephone; % of household with female head of household |
| | Race | 3 | % of population who are white, black, or have two or more races |
| | Marital status | 6 | % of male/female never married; % of male/female widowed; % male/female divorced; |
| | Housing cost | 4 | % of housing as of total income with/without mortgage; median owner's housing cost with/without mortgage |
| | Language spoken | 1 | % of population who are English-spoken only |
| | Disability | 2 | % of population who have disability; % of population over 65 who have disability |
| | Veteran status | 2 | % of population who are veterans; % population over 65 who are veterans |
| | Gini | 1 | Gini index of inequality |
| **Socio-Economic Status** | Poverty | 1 | Poverty rate |
| | Income | 1 | Median income |
| | Education | 2 | % of population with at least high school degree, or at least bachelor's degree |
| **Occupation** | Occupation | 4 | % of workers who are private worker, government worker, self-employed worker, or unpaid family worker |
| | Commute to work | 4 | % of population who commute via car/truck/van, carpool, public transportation, or walking |
| **Health Insurance** | Health insurance coverage | 2 | % of population who do not have health insurance; % of population over 65% who do not have health insurance |

## F. Computable Phenotyping for SDoH Stratification

To identify distinct patient groups with different SDoH characteristics, we experimented with two unsupervised clustering algorithms, KMean clustering and Hierarchical clustering. KMeans clustering aims to partition data into pre-defined k number of clusters in which each observation belongs to the cluster with the nearest mean. It iteratively assigns data points to clusters and updates the cluster centroid until convergence. While hierarchical clustering creates a hierarchy of clusters by either merging smaller clusters into larger ones (bottom-up approach) or dividing large clusters into smaller ones (top-down approach) based on a similarity measure.

We evaluated the performance of the two algorithms by varying the pre-defined number of clusters from 2 to 6. The optimal number of clusters and the best-performing algorithm were determined using the silhouette score, which measures the cohesion within clusters and the separation between clusters. A higher silhouette score indicates better-defined clusters, with scores ranging from -1 to 1. The clustering algorithm and the number of pre-defined clusters that achieved the highest silhouette score were selected for subsequent analysis, as they are expected to provide the most meaningful and well-separated clusters based on patients' SDoH characteristics. Each zip code was then associated with one of the identified clusters. To link the assigned clusters to the OneFL+ cohort, we matched the 5-digit zip codes between the two datasets to characterize patients in the OneFL+ cohort based on their corresponding SDoH cluster.

## G. Statistical Analysis

We first calculated the crude odds ratio to examine the unadjusted association between the identified SDoH clusters and predicted pre-operative frailty in the OneFL+ cohort. The Chi-square test was used to evaluate the statistical significance of differences. To account for potential demographic confounding factors, we used logistic regression model adjusting for age, sex, race, and the number of comorbidities, which allows us to determine the independent association between the identified SDoH clusters and predicted pre-operative frailty after controlling for those patients' characteristics. Statistical analyses were performed using R version 4.2.1. Figure 1 graphically illustrates the overall study design.

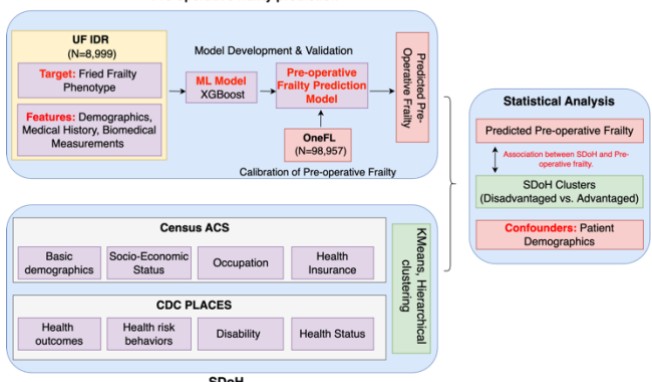

Fig. 1.    Graphical illustration of the overall study Design.

A total of 880 zip codes with complete SDoH features were included in the studies, covering 98,957 patients. The XGBoost model built using UF IDR data achieved a mean Area Under the Receiver Operating Characteristic Curve (AUC) of 0.74 (SD 0.01), sensitivity of 0.63 (SD 0.03), and specificity of 0.72 (SD 0.01), in predicting pre-operative frailty which is comparable to the previous attempts in using EHR to predict frailty in different settings [20], [21]. Based on the selected cut-point, we identified 20,685 (20.9%) frail patients in the OneFL+ cohort. Compared to their non-frail counterparts, the predicted frail patients were significantly older, with lower mean hematocrit, hemoglobin, and platelet count, but higher number of comorbidities. Table 2 compares the characteristics of predicted frail and non-frail patients in the OneFL+ cohort.

TABLE II.    COMPARISON OF KEY CHARACTERISTICS BETWEEN PREDICTED FRAIL AND NON-FRAIL PATIENTS IN THE ONEFL+ COHORT.

| Features | Frail (N=20685) | Non-Frail (N=78272) | p-value |
|---|---|---|---|
| Age | 79.8 (7.8) | 74.0 (6.4) | < 0.001 |
| Hematocrit | 29.5 (4.8) | 34.4 (6.1) | < 0.001 |
| Hemoglobin | 9.5 (1.5) | 11.2 (2.0) | < 0.001 |
| Platelet count | 208.1 (105.8) | 211.3 (87.5) | < 0.001 |
| Number of comorbidities | 30.1 (13.1) | 13.2 (9.4) | < 0.001 |
| Gender | | | < 0.001 |
| Male | 8860 (17.9%) | 40549 (82.1%) | |
| Female | 11824 (23.9%) | 37712 (76.1%) | |
| Race | | | < 0.001 |
| White | 15795 (20.2%) | 62445 (79.8%) | |
| Others | 4890 (23.6%) | 15827 (76.4%) | |

KMeans clustering outperformed Hierarchical clustering in identifying SDoH clusters uniformly when the number of clusters was pre-defined from 2 to 6, as shown in Table 3. Specifically, KMeans clustering with 2 clusters achieved the highest silhouette score of 0.17. Therefore, subsequent analyses were based on the 2 clusters identified by the KMeans clustering algorithm.

TABLE III.    COMPARISON OF PERFORMANCE IN IDENTIFYING SDoH CLUSTERS USING KMEANS AND HIERARCHICAL CLUSTERING WITH DIFFERENT NUMBERS OF CLUSTERS. A HIGHER SILHOUETTE SCORE INDICATES BETTER-DEFINED CLUSTERS, WITH SCORES RANGING FROM -1 TO 1.

| Number of clusters | Silhouette score (KMeans) | Silhouette score (Hierarchical clustering) |
|---|---|---|
| 2 | **0.17** | 0.13 |
| 3 | 0.15 | 0.13 |
| 4 | 0.16 | 0.13 |
| 5 | 0.14 | 0.13 |
| 6 | 0.12 | 0.10 |

Most SDoH characteristics differed significantly (p < 0.05) between the two identified clusters. Cluster 0 had better health conditions (e.g., lower disability rate, lower prevalence of chronic diseases such as arthritis, cancer, chronic kidney disease among adults aged 18 and above), better economic status (e.g., higher median income, higher housing expense, and lower poverty rate), higher education level (e.g., higher percentage of at least Bachelor's degree), and lower rates of no leisure time physical activity than cluster 1. Table 4 shows the comparison of SDoH features between the advantaged

and disadvantaged clusters. Given the relatively better SDoH profile, we considered cluster 0 the advantaged SDoH cluster and used it as the reference group in the subsequent statistical analysis.

TABLE IV.  COMPARISON OF SDOH FEATURES BETWEEN THE 2 CLUSTERS IDENTIFIED BY THE KMEANS CLUSTERING ALGORITHM.

| Feature | Cluster 1 | Cluster 0 | P |
|---|---|---|---|
| % arthritis among adults aged >=18 years | 27.65 (4.92) | 23.27 (5.63) | <0.01 |
| % binge drinking among adults aged >=18 years | 14.26 (1.83) | 15.31 (2.30) | <0.01 |
| % cancer (excluding skin cancer) among adults aged >=18 years | 7.15 (2.00) | 7.23 (2.44) | 0.46 |
| % chronic kidney disease among adults aged >=18 years | 3.75 (0.63) | 2.96 (0.66) | <0.01 |
| % chronic obstructive pulmonary disease among adults aged >=18 years | 10.24 (2.08) | 6.93 (1.69) | <0.01 |
| % coronary heart disease among adults aged >=18 years | 8.3 (1.91) | 6.54 (1.99) | <0.01 |
| % current asthma among adults aged >=18 years | 9.4 (0.78) | 8.12 (0.58) | <0.01 |
| % current lack of health insurance among adults aged 18-64 years | 24.89 (5.63) | 18.96 (5.05) | <0.01 |
| % current smoking among adults aged >=18 years | 22.11 (3.46) | 15.0 (2.85) | <0.01 |
| % depression among adults aged >=18 years | 19.72 (1.86) | 17.61 (1.81) | <0.01 |
| % diagnosed diabetes among adults aged >=18 years | 13.39 (2.19) | 10.17 ( 1.88) | <0.01 |
| % fair or poor self-rated health status among adults aged >=18 years | 21.1 (3.69) | 13.95 (2.47) | <0.01 |
| % mental health not good for >=14 days among adults aged >=18 years | 16.45 (1.65) | 13.98 (1.79) | <0.01 |
| % no leisure-time physical activity among adults aged >=18 years | 31.06 (4.23) | 23.02 (3.59) | <0.01 |
| % obesity among adults aged >=18 years | 34.75 (3.76) | 27.58 (3.06) | <0.01 |
| % physical health not good for >=14 days among adults aged >=18 years | 14.09 (1.74) | 10.31 (1.35) | <0.01 |
| % stroke among adults aged >=18 years | 4.31 (0.84) | 3.01 (0.77) | <0.01 |
| % having visits to doctor for routine checkup within the past year among adults aged >=18 years | 75.83 (2.67) | 74.9 (3.26) | <0.01 |
| Gini index | 0.43 (0.04) | 0.45 (0.06) | <0.01 |
| Median income | 66192.16 (12986.97) | 106325.51 (27503.01) | <0.01 |
| Median owner housing cost (with mortgage) | 1409.85 (263.68) | 2223.73 (635.43) | <0.01 |
| Median owner housing cost (without mortgage) | 451.22 (104.94) | 793.47 (303.98) | <0.01 |
| % no telephone for 65 and above | 0.02 (0.03) | 0.01 (0.01) | <0.01 |
| % 65 and over with no health insurance | 0.01 (0.01) | 0.01 (0.02) | 0.41 |
| % at least Bachelor's degree | 0.20 (0.07) | 0.43 (0.12) | <0.01 |
| % at least high school degree | 0.85 (0.07) | 0.93 (0.04) | <0.01 |
| % black | 0.18 (0.19) | 0.08 (0.07) | <0.01 |
| % female divorced | 0.15 (0.04) | 0.13 (0.04) | <0.01 |
| % female never married | 0.28 (0.10) | 0.25 (0.10) | <0.01 |
| % female widowed | 0.11 (0.04) | 0.10 (0.04) | <0.01 |
| % housing as income (with mortgage) | 23.04 (3.92) | 23.31 (3.83) | 0.60 |
| % housing as income (without mortage) | 10.71 (2.96) | 11.82 (3.47) | <0.01 |
| % male divorced | 0.13 (0.04) | 0.10 (0.03) | <0.01 |
| % male never married | 0.35 (0.11) | 0.30 (0.1) | <0.01 |
| % male widowed | 0.04 (0.02) | 0.03 (0.02) | <0.01 |
| % population no health insurance | 0.07 (0.05) | 0.05 (0.04) | <0.01 |
| % private worker | 0.73 (0.08) | 0.75 (0.06) | <0.01 |
| % self-employed worker | 0.06 (0.04) | 0.07 (0.03) | 0.03 |
| % spoken only English | 0.80 (0.19) | 0.74 (0.21) | <0.01 |
| % two or more races | 0.09 (0.07) | 0.12 (0.08) | <0.01 |
| % unpaid family worker | 0.00 (0.01) | 0.00 (0.0) | 0.82 |
| % white | 0.66 (0.21) | 0.73 (0.14) | <0.01 |
| % carpooled for work | 0.11 (0.05) | 0.09 (0.03) | <0.01 |
| % car/truck/van for work | 0.95 (0.04) | 0.94 (0.06) | 0.88 |
| % government worker | 0.14 (0.07) | 0.11 (0.05) | <0.01 |
| % public transportation for work | 0.01 (0.02) | 0.01 (0.02) | 0.55 |
| % walking for work | 0.01 (0.02) | 0.02 (0.03) | 0.67 |
| Poverty rate | 0.17 (0.07) | 0.09 (0.05) | <0.01 |
| Average household size | 2.48 (0.29) | 2.42 (0.37) | 0.01 |
| Disability rate | 0.20 (0.06) | 0.14 (0.04) | <0.01 |
| Disability rate over 65 | 0.37 (0.08) | 0.29 (0.07) | <0.01 |
| % female household | 0.22 (0.10) | 0.14 (0.07) | <0.01 |
| % living alone | 0.29 (0.08) | 0.27 (0.1) | <0.01 |
| Veterans rate | 0.09 (0.04) | 0.08 (0.04) | <0.01 |
| Veterans rate over 65 | 0.18 (0.07) | 0.16 (0.06) | <0.01 |

The odds of being frail were higher for individuals from the disadvantaged cluster across all models, though the strength of the association varied, as shown in Table 5. In the unadjusted model, individuals from the disadvantaged SDoH cluster had 1.13 times higher odds of being frail (95% CI 1.10-1.67) compared to individuals from the advantaged SDoH cluster. After adjusting for age, sex, and race, the odds ratio increased to 1.32 (95% CI 1.28-1.36). With additional adjustment for the number of comorbidities, the odds ratio lowered to 1.21 (95% CI 1.16-1.26). Our study revealed that, there is evidence that exposure to adverse SDoH was independently associated with increased likelihood of being frail even after controlling for potential confounders.

TABLE V.    THE ASSOCIATION BETWEEN SDoH CLUSTERS AND PRE-OPERATIVE FRAILTY RISK AFTER ADJUSTING POTENTIAL CONFOUNDERS.

| Models | Odds ratio | 95% CI |
|---|---|---|
| Unadjusted | 1.13 | [1.10, 1.67] |
| Adjusted for age, sex, and race | 1.32 | [1.28, 1.36] |
| Adjust for age, sex, race, and number of comorbidities | 1.21 | [1.16, 1.26] |

## DISCUSSION

This study investigated the impact of community-level SDoH on pre-operative frailty in a diverse population of surgical patients in Florida. By leveraging a large-scale and data-driven approach, we uncovered the interaction between community-level SDoH and pre-operative frailty risk. Our findings suggest that individuals living in communities with adverse SDoH profiles, characterized by high disability rates, lower economic status, and poorer health conditions, have a higher likelihood of being pre-operatively frail, even after adjusting for potential confounders such as age, sex, race, and number of comorbidities.

The association between adverse SDoH and increased frailty risk aligns with previous studies that examined the impact of individual-level SDoH on frailty. Lang et al. found that poorer individual wealth status (estimated using financial and housing wealth and other assets) was independently associated with a higher risk of frailty in community-dwelling older adults [17]. Similarly, previous studies also demonstrated that older adults with lower education levels and income, and higher number of chronic conditions were more likely to be frail [15], [22], [23]. Our study extends these findings by demonstrating the influence of community-level SDoH on pre-operative frailty, highlighting the importance of considering the broader social and environmental context in which patients live when assessing frailty risk.

The identification of two distinct SDoH clusters with significant differences in health conditions, economic status, and education levels using zip code-level characteristics, underscores the heterogeneity of communities in Florida and the potential impact of these disparities on health outcomes. The disadvantaged SDoH cluster was characterized by higher rates of chronic disease, lower median income, and lower education attainment compared to the advantaged SDoH cluster. This finding aligns with the growing evidence that highlights the influence of social and environmental factors on individuals' health outcomes. Individuals living in disadvantaged communities often face multiple barriers to maintaining good health, such as limited access to healthy food options, safe spaces and available time for physical activity, and quality healthcare services[24], [25], [26]. Over time, these social and environmental challenges can contribute to a higher risk of adverse health outcomes, such as diabetes [26], [27], cardiovascular disease [28], [29], [30], and cancer [29], [30].

The association between SDoH clusters and the risk of pre-operative frailty was still statistically significant even after adjusting for potential confounders, suggesting that community-level SDoH may have an independent effect on pre-operative frailty. This finding has important implications for clinical practice and public health interventions. Clinicians should consider the social and environmental context of their patients when developing pre-operative care plans. Public health interventions aimed at improving community-level SDoH, such as increasing access to healthcare and promoting education may help mitigate frailty risk and eventually improve surgical outcomes in vulnerable populations.

There are some limitations to be noted. First, although we considered various community-level SDoH, including area demographics, socio-economic status, and health conditions, our analysis did not encompass the full scope of SDoH factors, possibly omitting other influential elements. Second, we incorporated SDoH at the zip code level, which may not capture the finer characteristics of individuals' immediate living environment. While zip code-level data provides valuable insights into the general socioeconomic and health conditions for a community, it may not account for the heterogeneity within the area or the specific neighborhood-level factors that directly impact an individual's health outcome. Future studies could utilize more fine-grained SDoH data, such as census tract or block group level information, to better understand the localized influence on frailty. Third, the cross-sectional design of the study precludes the causal inference between SDoH and frailty. Finally, the generalizability of our findings to other regions, populations, and settings may be limited and needs further investigation. Despite these limitations, our study offers a crucial step toward understanding the complex interactions between community-level SDoH and pre-operative frailty.

## CONCLUSION

This study demonstrates that the adverse SDoH profile is independently associated with higher pre-operative frailty risk in a diverse population of surgical patients in Florida. Our findings highlight the potential for public health policies and clinical practice to consider patients' social and environmental context to improve pre-operative care and surgical outcomes.

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
