# OpenReview forum: "Elucidating The Impact of Community-Level Social Determinants of Health on Pre-operative Frailty: A Data-Driven Study in Florida"
_IEEE.org/EMBS/BHI/2024/Conference — IEEE BHI'24_

### Official Review · Reviewer_zkgU · 2024-07-31
**The authors applied an unsupervised learning analysis on a dataset to evaluate the impact of community-level social determinants of health  on pre-operative frailty**

**Overall Rating:** 7
**Confidence:** 4

**Other Quality Metrics:**

(a) Clarity of writing; Great
(b) Clinical Significance; Great
(c) Methodological Novelty; Fair/Good
(d) Experiments and Results; Great

**Questions For The Authors:**

-	It is unclear what the reason is to use XGBoost in the analysis involving unsupervised learning. Was the model used to infer the possible frailty labels in the OneFL+ cohort after being trained on the UF IDR dataset? Why would the authors do that if the Fried frailty phenotype could be assumed based on the different criteria mentioned? What is the methodology used for training this model and what is the performance of this model?
-	Why the authors didn’t use the elbow method to verify the correct number of clusters chosen?
-	Fig 1 is too small to be readable

**Strengths:**

This paper is an interesting work

**Summary Of The Paper:**

The authors applied an unsupervised learning analysis on a dataset to evaluate the impact of community-level social determinants of health  on pre-operative frailty

**Weaknesses:**

This paper requires some clarifications (see below)

---

### Official Review · Reviewer_Yd26 · 2024-08-10
**Exploring the Impact of Community-Level Social Determinants of Health on Pre-operative Frailty in Florida**

**Overall Rating:** 7
**Confidence:** 3

**Other Quality Metrics:**

(a) Clarity of writing: good
(b) Clinical Significance: good
(c) Methodological Novelty: good
(d) Experiments and Results: fair

**Questions For The Authors:**

Could the prediction accuracy of model be improved by incorporating additional variables, such as individual behavioral factors ?

**Strengths:**

This study effectively utilizes machine learning techniques, including eXtreme Gradient Boosting (XGBoost), to predict pre-operative frailty, showcasing a modern approach to healthcare research. Leveraging data from the OneFlorida+ Consortium, which encompasses a wide and diverse population, enhances the generalizability of the findings. Additionally, the research extends beyond individual factors by exploring broader community-level Social Determinants of Health (SDoH), offering a more comprehensive perspective on the factors influencing pre-operative frailty.

**Summary Of The Paper:**

The paper investigates the relationship between community-level social determinants of health (SDoH) and pre-operative frailty among a diverse population in Florida. The study utilized machine learning models to predict pre-operative frailty using data from the University of Florida (UF IDR) and the OneFlorida+ Clinical Research Consortium. The research identifies significant associations between adverse SDoH profiles and higher pre-operative frailty risk, emphasizing the importance of considering patients' social environments in pre-operative care to improve surgical outcomes.

**Weaknesses:**

Despite using a large dataset, focusing on Florida limits the generalizability of its findings to regions with different demographic and socioeconomic profiles.

---

### Official Review · Reviewer_iNnN · 2024-08-19
**Review of paper 274**

**Overall Rating:** 7
**Confidence:** 3

**Other Quality Metrics:**

a) good
b) great
c) fair
d) good

**Questions For The Authors:**

Although there are significant weaknesses which hinder the wider applicability of this study, it is a good preliminary study to drive the discussion and research on equitable and fair clinical practice across subgroups of the population.

**Strengths:**

- The study leverages a large and diverse dataset enhancing the generalizability of the findings.
- The authors utilize a machine learning model to predict pre-operative frailty, considering population characteristics and diversities, which is a novel approach in this context.

**Summary Of The Paper:**

The authors developed a machine learning model to predict pre-operative frailty using an institutional dataset and applied it to a more geographically diverse population from the OneFlorida+ Clinical Research Consortium. They identified distinct patient profiles based on zip code-level SDoH characteristics. They evaluated a multivariate logistic regression model to examine the association between
SDoH profiles and pre-operative frailty risk, concluding that considering patients’ social context could improve pre-operative care and surgical outcomes, informing clinical practice and policies. Overall, the paper makes a valuable contribution to understanding the impact of community-level SDoH on pre-operative frailty, but it also highlights areas for further research and improvement.

**Weaknesses:**

- The study finds that adverse community-level SDoH profiles are independently associated with higher pre-operative frailty risk. However, causal analysis is necessary to ensure that the analysis is not biased with respect to certain subgroups. This specific design limits the ability to infer causality between SDoH and frailty.
- The study relies on zip code-level SDoH data, which may not capture finer details of individuals’ immediate living environments.
- The findings may not be generalizable to regions outside Florida or to different populations and settings.
- Although the study adjusts for several potential confounders, there may still be unmeasured variables that could influence the results.

---

### Decision · Program_Chairs · 2024-09-23

Accept